# Food Consumption, Nutrient Intake and Status during the First 1000 Days of Life in The Netherlands: A Systematic Review

**DOI:** 10.3390/nu11040860

**Published:** 2019-04-16

**Authors:** Sovianne ter Borg, Nynke Koopman, Janneke Verkaik-Kloosterman

**Affiliations:** National Institute for Public Health and the Environment, 3721 MA Bilthoven, The Netherlands; sovianne.ter.borg@rivm.nl (S.t.B.); nynke.koopman@rivm.nl (N.K.)

**Keywords:** first 1000 days, nutritional assessment, Netherlands, early life, nutritional status, pregnancy, lactation

## Abstract

Adequate nutrition is essential for growth and development in early life. Nutritional data serves as a basis for national nutritional guidelines and policies. Currently, there is no insight into the availability of such data during the first 1000 days of life. Therefore, a systematic review was performed, following the PRISMA reporting guideline, to identify studies on food consumption, nutrient intake or status in the Netherlands. Potential gaps were identified, and the quality of the studies is discussed. The databases Embase and Medline were used, as well as databases from national institutes. Articles published in 2008–2018 were screened by two independent reviewers. In total 601 articles were identified, of which 173 were included. For pregnant women, 32 studies were available with nutritional data, for young children 40 studies were identified. No studies were available for breastfeeding women. A large variety of foods and nutrients were assessed, however certain nutrients were lacking (e.g., vitamin K). Overall, the studies had methodological limitations, making the data unsuitable to assess nutrient inadequacies. There is a need for recent, high quality nutritional research to strengthen the understanding of the nutritional needs and deficiencies during early life, and is fundamental for national guidelines and policies.

## 1. Introduction

Adequate nutrition is essential for growth and development in early life [1]. Sufficient energy and nutrient intakes during pregnancy are needed to support maternal tissue growth, as well as to prevent unfavorable birth outcomes. A well-known example is the relation between folic acid intake of the mother and neural tube defects in the offspring [2]. In addition to these relative short-term effects, there are long-term effects of prenatal nutrient exposure. Evidence is growing that early exposure, during pregnancy or even periconceptionally, can affect the metabolism of the child. It is suggested that maternal pre-pregnancy obesity is associated with offspring adiposity, potentially through leptin epigenetic regulations [3,4]. Results from the Dutch famine birth cohort study show an association between severe prenatal malnutrition and the development of obesity and coronary heart disease later in life [5]. This relation between the metabolic programming in early life, and the development of non-communicable diseases in later life, is known as the ‘Developmental Origins of Health and Disease (DOHaD) hypothesis’ [6,7,8].

Although there is little concern for any energy insufficiency in affluent countries, pregnant and breastfeeding women may be at risk of certain nutrient deficiencies, due to an increased nutritional need, and low nutrient-dense food patterns. Previous research indicates that pregnant women, living in affluent countries, are at risk of folate, iodine, iron and vitamin D inadequacies [9,10].

In addition, excess nutrient intakes are a potential concern. High maternal intakes of vitamin A for instance can cause hepatotoxicity and birth defects [11].

To meet the nutritional needs, specific nutritional guidelines are set for pregnant women, lactating women and their offspring. The European Food Safety Authority recommends, amongst others, an increased intake of folate and iodine for pregnant and lactating women [12]. The EURRECA (the European Micronutrient Recommendations Aligned) Network identified differences in certain micronutrient recommendations for pregnant women among European countries [13]. Both the EFSA and EURRECA indicate the difficulties concerning data variability, interpretation and the absence of certain data, such as for vitamin D status in breast feeding women [12,13]. Recently, consensus recommendations were published by the Early Nutrition Project [14], aimed at aligning international recommendations for women and children in affluent countries. They emphasize the importance of increasing the dietary quality, rather than the quantity. They advise folic acid supplementation before conception and in early pregnancy, and the supplementation of iron, vitamin D, vitamin B_12_ and iodine in case of deficiencies. As EFSA and EURRECA, the Early Nutrition Project identified certain data gaps: The impact of nutrition during breastfeeding on the child’s later health, and optimal weight gain during early pregnancy.

In 2008, the Dutch Health Council published guidelines on vitamin A, folic acid and vitamin D for the general population, including pregnant women and breastfeeding women [15,16,17,18]. Pregnant women are advised to not use supplements containing vitamin A (as retinol) and limit the consumption of liver [16]. In addition, they are advised to use a folic acid supplement (400 µg per day), starting four weeks prior to conception until eight weeks after conception [15]. Pregnant women are also advised to use a vitamin D supplement (10 µg per day) [19]. For breastfeeding women, no specific micronutrient guidelines are in place. Vitamin K supplementation is recommended for newborns (1 mg at birth and 150 µg per day for breastfed infants, 8 days to 12 weeks after birth), as well as vitamin D supplementation (10 µg per day, up to 4 years of age) [19,20]. In addition to the micronutrient guidelines, there are guidelines on food safety, for instance to prevent contamination with *Listeria* or excessive intakes of caffeine during pregnancy [21]. For breastfeeding women, it is advised to consume adequate amounts of water (2–2.5 L per day) [22].

The Dutch Health Council is currently re-evaluating the nutritional guidelines for the first 1000 days of life, and will publish specific national guidelines for pregnant women, breastfeeding women and the child up to two years of age [23].

Data on nutrient intake and status during the first 1000 days of life are used to study the relations between nutrition and health, and to develop strategies to prevent chronic diseases later on in life. Nutritional data are also used to assess potential inadequacies and toxicities, and as such are important for nutritional policy. In addition, nutritional data can be used to calculate the optimal level of food fortification, and to monitor the impact of such a public policy over time.

Currently, there is however no insight on the availability of such nutritional data. Therefore, the aim of the present systematic review is to identify studies which assessed the food consumption, nutrient intake, or biochemical nutrient status during the first 1000 days of life in the Netherlands. Possible data gaps were identified, and the quality of the available studies is discussed.

## 2. Method

This systematic review is reported following the Preferred Reporting Items for Systematic reviews and Meta-Analyses (PRISMA) guideline [24].

The electronic database Embase (Embase.com) was used, which included both the Embase as well as Medline database. Studies published between January 2008 up to May 2018 were included. Scientific posters and abstracts were not included. The Dutch Health Council is currently re-evaluating the nutritional guidelines [23]. In previous guidelines on micronutrients, they included literature up to 2008 [15,16,17,18]. Therefore, for the current systematic review, 2008 was chosen as the starting date for the literature search, to identify more recently-published studies.

A PICO model was used to formulate the search strategy [25]. The population (P) was defined as: First 1000 days, pregnant women, mothers during the breastfeeding period and children up to two years of age. Populations living in the Netherlands without medical illnesses were included. Intervention (I) studies were excluded, except when baseline data was available, prior to the intervention. A comparison to a control group (C) was not taken into account, as we did not study an intervention. However, if nutritional data was available from a healthy control group, these data were included. The outcomes (O) of interest were data on food and nutrient intake, dietary supplement use and biochemical nutrient intake markers. Emtree index terms were used and exploded to find as many relevant studies as possible. The search filters: Humans, publication data and publication type, were used. No language restrictions were applied. The search string is provided in the Appendix A
Table A1. In addition, reports from the National institute of food safety (RIKILT), the National Institute for Public Health and the Environment (RIVM) and the Netherlands Organization for applied scientific research (TNO) were identified through the institute-websites [26,27,28]. For the identification and selection of relevant reports, the same PICO and exclusion criteria were used as for the scientific articles.

Titles and abstracts were screened independently by two reviewers (N.K., S.t.B.). Predefined exclusion criteria were used: Studies published before 2008; did not contain Dutch data; included a population with a medical illness or premature infants; the population was not pregnant, breastfeeding, or had a mean age above two years; preconception data; no data on food or nutrient intake, supplement use or nutrient status; intervention studies without reporting baseline data; case studies. Subsequently full texts were retrieved and assessed based on the selection criteria stated above. If the full text could not be retrieved, they were excluded. Disagreement on the inclusion of a study was resolved through consensus or the consultation of a third reviewer (J.V.-K.). The flow diagram of the study selection is provided in Figure 1.

For each of the included studies the following data was extracted by the two reviewers (NK, StB): Study name, year of publication and years of data collection, type of study (cohort, case-control, randomized controlled trial), study population characteristics (i.e., location, gestational age and birth weight, age, ethnicity, BMI), supplement intake data, dietary assessment method and validation, reported foods, nutrients and biochemical markers of nutrient intake status. Data on breastfeeding was considered as a food intake of children, although the assessment took place among their mothers. For data interpretation, articles referring to the same study or cohort name were grouped.

The reference management software EndNote X9 was used during the selection procedure. The study characteristics were recorded in Microsoft Excel (Microsoft Office Professional Plus 2010).

## 3. Results

In total 601 articles were identified, of which 173 met the inclusion criteria (Figure 1). The main reasons for exclusion were the absence of nutritional data, and a non-relevant study population. Of the 173 articles, 109 articles were identified with nutritional data on pregnant women, zero articles with data on breastfeeding women, and 114 articles with data on children up to the age of two years. Certain articles contained information on both pregnant women and young children. The study characteristics can be found in Table 1 for pregnant women and Table 2 for children up to two years of age respectively. The cohorts with the largest number of publications were the Generation R study (*n* = 78), the ABCD study (*n* = 17), KOALA study (*n* = 7) and PIAMA study (*n* = 6). Publications with data on pregnant women originated from 32 different studies. For children up to two years of age, 40 different studies were identified. The number of study participants ranged from 8 to 7857 pregnant women, and 9 to 7857 children.

### 3.1. Food Consumption

For pregnant women, the most frequently reported data, on food consumption, were on alcoholic beverages (*n* = 18), fish (*n* = 5), fruit, and vegetable consumption (both *n* = 4) (Table 3). Alcohol exposure (yes/no) during pregnancy was assessed, and to a lesser extend the amount which was consumed. Although there was a large variety in the foods that were assessed, these originated mainly from one single cohort (i.e., Generation R study). The most frequently reported data for children up to two years of age was on feeding practices (i.e., breastfeeding, formula feeding) (Table 3). Whether children were breastfed was reported in 28 different studies, and the period of breastfeeding was also frequently assessed. Formula feeding was reported in 12 different studies.

### 3.2. Nutrient Intake

Maternal energy and macronutrient intake were most frequently stated, in up to five different studies (Table 4). For the vitamins, data was reported for retinol, riboflavin, nicotinamide and folate. These data originated from one single cohort (i.e., Generation R). No data was available for beta-carotene, thiamine, pantothenic acid, pyridoxine, biotin, cobalamin and vitamins C, D, E and K. For the maternal mineral intake, data was available for calcium, iron, magnesium, phosphorus and sodium. As with vitamins, the mineral intake data originated from one single cohort (i.e., Generation R). No data was available for chlorine, chromium, copper, iodine, manganese, molybdenum, potassium, selenium and zinc intake. Two studies reported on caffeine intake (i.e., Generation R and ABCD study). For children up to two years, primarily energy and the macronutrients were reported, by four and three different studies, respectively. A large variety of vitamins and minerals were assessed, however originating from one or two different studies (i.e., Generation R and Eat complete test). Data was reported for: Thiamine, riboflavin, niacin, vitamin A, beta-carotene, pyridoxine, cobalamin, folate, vitamins C, D and E, calcium, copper, iodine, iron, magnesium, phosphorus, potassium, selenium, sodium and zinc. In addition, data was available on water intake. No data was available for pantothenic acid, biotin, vitamin K, chlorine, chromium, manganese and molybdenum.

### 3.3. Biochemical Nutrient Status

Most frequently stated maternal nutrient status markers were fatty acids and vitamin D, which were reported by six and four different studies, respectively (Table 5). The B vitamins (folate, cobalamin) and homocysteine were stated by two to three studies. Retinol, ferritin, iodine, and iron were reported in one study each. No data for vitamin C, calcium, copper, selenium and zinc status was available. For children up to two years of age, nutrient status data was available for vitamin D (three different studies), fatty acids (one study) and ferritin (three different studies) (Table 5). For calcium, one study provided the data. No data was available for the B vitamins, vitamin C, copper, iodine, iron, selenium and zinc status.

### 3.4. Supplement Use

The most frequently reported supplement was folic acid, being about whether women used this supplement during their pregnancy (thirteen different studies, Table 6). Several studies reported on the duration and the period (e.g., preconception) of folic acid supplement use. For children, the most often assessed supplement was vitamin D, which was reported by ten different studies (Table 6). 

### 3.5. Time Span

The years in which the data were collected ranged from 1989 until 2016 (see Table 3, Table 4, Table 5 and Table 6). For pregnant women, the most recent data originated from 2016 (alcohol consumption), however most data was collected before 2006. Nutrient status data was more recent, with assessments of vitamin B_12_, folate and homocysteine up to 2014. For children up to two years of age, most food consumption and nutrient intake data were collected before 2014. The most recent nutrient status data originated from 2014 (ferritin), the other status markers (such as vitamin D) were assessed before 2009.

### 3.6. Dietary Assessment Methods

For the assessment of alcohol use, breastfeeding practices and folic acid supplement use, general questionnaires were used. Food consumption and nutrient intake were mainly assessed via a semi-quantitative food frequency questionnaire (FFQ) (see Table 3 and Table 4). In addition, a two day food record was used by five different studies, and a dietary recall was performed by three studies (of which one performed a repeated dietary recall). There was one study which used a semi-weighted food record, and one study which used duplicate portions. Often no information on any validation of the methodology was provided. In case the method was validated, it was generally used after adaptations, and not in its original form. Only one study explicitly reported the validation of the FFQ among pregnant women.

## 4. Discussion

To our knowledge, this review is the first to provide a comprehensive overview on Dutch nutritional data, and to include the full population-range of the first 1000 days of life. In addition, nutritional intake data as well as biochemical nutrient status were included in this review, which are essential for determining potential nutrient deficiencies or excesses.

### 4.1. Previous Findings

Previously, a systematic review was published on nutrient intake and biochemical nutrient status among pregnant women in affluent countries [9], and more recently on pregnant adolescents [202]. The first review however included one Dutch study, performed in 1988. The second review, on pregnant adolescents, didn’t include data from the Netherlands. Although we identified data for pregnant women and young children, no data was found for breastfeeding women living in the Netherlands. There are however a few studies on breastfeeding women in affluent countries: A recent study assessed the dietary intake among French breastfeeding women, reporting their energy and macronutrient intake [203], and the energy and macronutrient of Greek breastfeeding women was assessed [204]. Other studies focus specifically on vitamin D intake and status [205,206]. Additional research is of interest, as adequate nutrition during this period within the 1st 1000 days (i.e., the breastfeeding period) contributes to maternal and infant health [207].

### 4.2. Strengths and Limitations

The present review provides a detailed overview of the available studies with nutritional data, there are however certain limitations that need to be addressed. Although the search string was developed with great care, we cannot guarantee that all relevant articles were identified. Unpublished data and publication bias may influence the findings, as well as the choice of search terms; however, as a large number of articles were identified, it is expected that our conclusions are robust. We identified multiple publications which referred to the same study. To prevent an overestimation of the available data, we grouped these references, assuming that these articles refer to an identical population. This assumption may not be correct for all articles, as subpopulations of a study may have been assessed, and published in separate articles. Studies which did not report a cohort or study name were assumed to be independent studies. These assumptions may affect the number of individual studies, which were available for the nutritional parameters.

### 4.3. Quality of the Studies

Although the results indicate that there is nutritional data available, the usability of the data, to evaluate the nutrient intake of pregnant women and children up to two years of age, is limited. Most of the identified data was from studies examining the association between maternal life styles, environmental determinants and health outcomes. Only a few studies were designed to evaluate the nutrient adequacy. Factors such as the time span, national representativeness and methodology affect the usability of the data, and will be discussed below.

This review includes data published since 2008. Although the most recent data originates from 2016, many of the assessments were performed prior to 2006. It can be questioned as to whether these data are still representative for the current dietary intake, as dietary patterns and food compositions may change over time [208]. In addition, changing national legislations and policies may have affected the nutrient composition of the food: In the Netherlands, fortification of foods with vitamin D and folic acid are allowed since 2007 [209]. In 2008 the Dutch policy on the addition of iodized salt changed, resulting in a decrease of iodine intake among the Dutch population [210].

Although the identified studies are dated, there are several cohort studies, which are currently running in the Netherlands. These may provide future data on pregnant women and their offspring. We are however unaware of ongoing studies among lactating women.

Most of the identified studies were not nationally representative for pregnant women or young children living in the Netherlands. Studies focused for instance on specific cities or areas within the Netherlands. The National Food Consumption Survey [144] contained recent and representative data (2012–2014) on young children (aged 1–3 years). Pregnant women and breastfeeding women were however excluded from this survey. 

A variety of dietary assessments were used. Most often articles stated that a general questionnaire was used, without further specifications. These questionnaires were utilised to give an impression of alcohol and dietary supplement use during the pregnancy. In addition, these questionnaires were employed to assess exclusive breastfeeding and the duration of breastfeeding. For dietary assessments, the food frequency questionnaire was most often used. FFQs are a cost-effective and time-saving method, with a low participant burden, which make them suitable for large epidemiological studies [211]. Although this method can be used to rank persons based on their intake, it often does not provide a valid estimate of the actual consumed amount, or the distribution of the (habitual) intake of the population, due to the generally limited number of items included in an FFQ [212,213]. This habitual intake distribution is needed to estimate potential deficiencies and excess intakes at population level, and can be obtained by repeated detailed data collection, e.g., dietary records or 24 h recalls corrected for the within-person variation. Although these methods have their own limitations, they perform better with respect to estimating the nutrient intake [214,215]. A few studies were identified which used two day dietary records, however these assessments were prior to 2006, or not in a national representative sample. Almost all studies relied on self-reported dietary intakes, which may influence the results. Only two studies used a (semi-) weighted assessment method.

Validation was often absent or poorly described in articles, and in some cases, the validation was performed in an inappropriate study population, such as older adults. One study mentioned that they used an FFQ that was validated in pregnant women [129]. The authors however recommend further validation, as they adjusted the questionnaire based on a pre-test by introducing plate photos.

Not all studies specified the nutrient database which was used for calculating the nutrient quantities. As these databases change over time, using a dated version may influence data quality.

### 4.4. Absence of Data for Certain Nutrients

For some nutrients, the intake or status was not assessed or it was scarce. This may be due to the study aim, focused on a single specific nutrient, or due to methodological constrains. For nutrient status, the absence of a good status marker, feasibility of the assessment method, or high costs, may be restricting factors, and might explain why certain data are scarce. 

Serum calcium for instance, is highly regulated within the body, and is strongly affected by other nutrients such as vitamin D, and so it has limited use for assessing calcium status [216]. This may explain why there was only one study found which assessed calcium status. Another example is iodine. Urinary iodine is a useful biomarker to assess the iodine intake status [217], it has however multiple disadvantages: A high burden on the study participants, high costs and possible incomplete 24 h urine collections. These factors may explain why it is not frequently included in studies. Our results identified only one study assessing iodine status, via a single spot urine sample, among pregnant women. Spot urine collection is less burdensome, however additional research is needed on its validation in pregnant women and children [218]. Spot urine uses an assumption regarding the 24 h urine volume, this assumption may however be incorrect, as a higher urinary volume was measured in the Dutch adult population [219].

### 4.5. Suggestions for Future Research

Overall, there is a lack of recent, representative nutritional data for pregnant women and young children. For some nutrients, and for breastfeeding women, the data were absent. There is a need for additional research: A future study should include a national representative population, and research is needed on breastfeeding women. In addition, in order to assess nutrient adequacy, the study should include a validated dietary assessment, such as a repeated 24-h dietary recall, combined with certain biochemical status assessments, especially for nutrients difficult to assess with, e.g., food consumption surveys. Examples are vitamin D, which is strongly influenced by sunlight exposure, and iodine, which is provided by fortified salt and is difficult to quantify through dietary assessment methods.

## 5. Conclusions

The current systematic review provides a comprehensive overview of the available nutritional data for the first 1000 days, assessed in the Netherlands. Although there was a large variety of foods and nutrients which were reported, most originated from one single study. No nutritional data was found for breastfeeding women, and the majority of assessments were performed before 2006. This time span, the methodology used and the absence of data for certain nutrients, make it difficult to evaluate the current nutritional intake and status of pregnant women, breastfeeding women and children up to two years of age. Overall, there is a need for high quality nutritional research during the full period of the first 1000 days of life. This data will strengthen the understanding of the nutritional needs and the potential nutrient deficiencies during early life, and is fundamental for national guidelines and policies.

## Figures and Tables

**Figure 1 nutrients-11-00860-f001:**
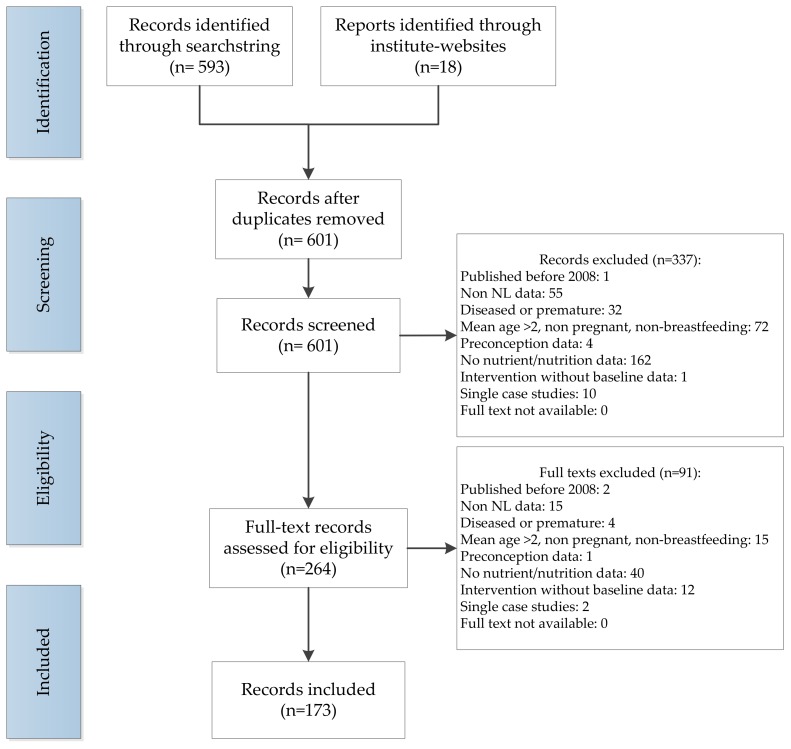
PRISMA flow diagram indicating the selection and inclusion procedure of articles with nutritional data assessed during the first 1000 days of life, in the Netherlands, published since 2008.

**Table 1 nutrients-11-00860-t001:** Overview of Dutch studies assessing food consumption, nutrient intake or biochemical nutrient status among pregnant women, published since 2008.

Study Name	Type of Study	Number of Articles	Year(s) of Data Collection	Number of Participants ^1^	Parameters Assessed				Method	
					Food Consumption	Supplement Intake	Nutrient Intake ^2^	Biochemical Nutrient Status	Dietary Assessment	Biochemical Nutrient Status
ABCD study	Cohort study	12 [29,30,31,32,33,34,35,36,37,38,39,40]	2003–2004	2274–4236	Alcohol, fish,	Folic acid, fish oil	Caffeine	25(OH)D, fatty acids, folate, vitamin B_12_	FFQ, questionnaire	Plasma, serum
DELIVER Study	Cohort study	2 [41,42]	2009–2011	5975	Alcohol, fruit, vegetables	Folic acid	-	-	Questionnaire	-
EUROCAT ^3^	Case-control study	1 [43]	1996–2005	3012	Alcohol	Folic acid	-	-	Questionnaire	-
Generation R Study	Cohort study	58 [40,44,45,46,47,48,49,50,51,52,53,54,55,56,57,58,59,60,61,62,63,64,65,66,67,68,69,70,71,72,73,74,75,76,77,78,79,80,81,82,83,84,85,86,87,88,89,90,91,92,93,94,95,96,97,98,99,100]	2001–2012	420–7857	Alcohol, bread, breakfast (cereals), condiments, dairy products, eggs, fat, fish, fruit, legumes, meat, non-alcoholic drinks, nuts and nut products, pasta, rice, potatoes, sauces, snacks, soup, soya and diet products, sweets, vegetables	Folic acid, (multi)vitamin	Energy, beta-carotene, caffeine, calcium, carbohydrate, fat, fiber, iron, magnesium, phosphorus, protein, sodium	25(OH)D, fatty acids, folate, iodine, homocysteine, vitamin B_12_	FFQ ^4^, questionnaire	Plasma, serum, urine
HAVEN study	Case-control study	2 [101,102]	2003	251–324	Alcohol	Folic acid	Energy, fat, folate, niacin, riboflavin	-	FFQ ^5^	-
Healthy pregnant	Case-control study	1 [103]	2004–2009	529	Alcohol	Folic acid	-	-	Questionnaire	-
HERNIA study	Case-control study	1 [104]	2006–2009	46	-	Folic acid	Energy, carbohydrates, fat, protein, retinol	Retinol	FFQ ^4^	Serum
IROSTAT Study	Case-control study	1 [105]	-	313	-	Iron	-	-	-	-
KOALA	Cohort study	7 [40,106,107,108,109,110,111]	2000–2002	521–2818	Alcohol	Vitamin D, multivitamin, fish oil	Energy, fat	25(OH)D, fatty acids	Questionnaire	Plasma
LINC	Cohort study	1 [112]	2011–2013	59	Alcohol	-	-	-	Questionnaire	-
MEFAB	Cohort study	2 [113,114]	1989–1995	292–1238	Alcohol	-	-	Fatty acids	Questionnaire	Plasma
Parents to Be	RCT	1 [115]	2003	1740	Alcohol	Folic acid	-	-	Questionnaire	-
‘Peiling melkvoeding van zuigelingen’	National survey	1 [116]	2015	1678	Alcohol	-	-	-	Questionnaire	-
PIAMA	Cohort study	3 [40,117,118]	1996–1997	3684–3963	Dairy products, eggs, fish, fruit, nuts and nut products, vegetables	Vitamin A, vitamin B complex, vitamin C, vitamin D, multivitamin, folic acid, calcium, iron	-	-	FFQ, questionnaire	-
Pregnancy Anxiety and Depression	Cohort study	2 [119,120]	2011–2013	2033–2287	Alcohol	-	-	-	Questionnaire	-
Rotterdam Predict Study	Cohort study	1 [121]	2009–2014	228	-	Folic acid	-	Folate, homocysteine	-	RBC, plasma
TRAILS	Cohort study	2 [122,123]	2001	679–1667	Alcohol	-	-	-	Questionnaire	-
Belderbos et al.	Cohort study	1 [124]	2006–2009	156	-	Vitamin D	-	-	Questionnaire	-
Diepeveen et al.	Case-control study	1 [125]	-	253	Alcohol	-	-	-	Questionnaire	-
Dirix et al.	Case-control study	1 [126]	-	90	Alcohol	-	-	-	Questionnaire	-
Doornbos et al.	RCT	1 [127]	-	36	Fish	-	-	Fatty acids	FFQ, questionnaire	Plasma
Lamb et al.	Cohort study	1 [128]	2008	323	Alcohol	-	-	-	Questionnaire	-
Merkx et al.	Cross-sectional study	1 [129]	2012	455	Fish, fruit, vegetables				FFQ ^3^	
Obermann-Borst et al.	Cohort study	1 [130]	2003–2007	120	-	Folic acid	-	-	Questionnaire	-
Oosterhoff et al.	Cross-sectional study	1 [131]	2008	8	Breastfeeding ^6^	-	-	-	Questionnaire	-
Poels et al.	Case-control study	1 [132]	2015–2016	283	Alcohol	Folic acid	-	-	Questionnaire	-
Savitri et al.	Cohort study	1 [133]	2010	130	Fasting	-	-	-	Questionnaire	-
Van Goor et al.	RCT	1 [134]	2004–2006	36	-	-	-	Fatty acids	-	RBC
Van Santen et al.	Cohort study	1 [135]	2009–2011	31	-	-	-	Ferritin, iron	-	serum
Vujkovic et al.	Case-control study	1 [136]	1999–2001	81	Alcohol	Folic acid	Energy	-	Questionnaire	-
Weernink et al.	Case-control study	1 [137]	2009–2010	548	Dairy	-	-	-	Questionnaire	-

^1^ as multiple references were available, the minimum and maximum number of the reported sample size are stated; ^2^ may include nutrient intake from supplements; ^3^ Northern Netherlands register on congenital anomalies; ^4^ dietary assessment method was reported as validated; ^5^ modified version of a validated questionnaire; ^6^ women’s perceptions of breastfeeding during the period of intention to breastfeed; 25(OH)D = 25-hydroxyvitamin D; FFQ = food frequency questionnaire; RBC = red blood cells; RCT = randomized controlled trial.

**Table 2 nutrients-11-00860-t002:** Overview of Dutch studies assessing food consumption, nutrient intake or biochemical nutrient status among children aged 0–2 years, published since 2008.

Study Name	Type of Study	Number of Articles	Year(s) of Data Collection	Number of Participants ^1^	Parameters Assessed				Method	
					Food Consumption	Supplement Intake	Nutrient Intake ^2^	Biochemical Nutrient Status	Dietary Assessment	Biochemical Nutrient Status
ABCD study	Cohort study	7 [30,34,138,139,140,141,142]	2003–2004	1459–3730	Breastfeeding, complementary feeding, formula, fruit, vegetables	-	-	-	Questionnaire	-
Bibo study	Case-control study	1 [143]	-	10	Breastfeeding, formula	-	-	-	Questionnaire	-
Dutch National Food Consumption Survey	National survey	1 [144]	2012–2014	517	Alcohol, breakfast (cereals), cakes and biscuits, composite dishes, condiments, confectionary, dairy products, eggs, fat, fish, fruit, legumes, meat, non-alcoholic drinks, nuts and nut products, potatoes, sauces, snacks, soup, vegetables	Vitamin D, folic acid, multivitamins, calcium	-	-	2 days recall & 2 days food record, questionnaire	-
Dutch EuroPrevall BCS	Cohort study	1 [145]	2006–2012	49	Breastfeeding	-	-	-	Questionnaire	-
‘Eet compleet test’	Cross-sectional	3 [146,147,148]	2011–2014	643–1526	Bread, breakfast (cereals), breastfeeding, cakes and biscuits, composite dishes, confectionery, dairy products, eggs, fat, fish, formula, fruit, legumes, meat, non-alcoholic drinks, nuts and nut products, potatoes, sauces, soup, soya and diet products, sweets, vegetables	Vitamin D	Energy, calcium, carbohydrates, copper, fat, fiber, folate, iodine, iron, magnesium, niacin, phosphorus, potassium, protein, retinol, riboflavin, selenium, sodium, thiamine, vitamin A, pyridoxine, cobalamin, vitamin C, vitamin D, vitamin E, water, zinc	-	2 days food record	-
GECKO study	Cohort study, RCT	2 [149,150]	2006–2007	65–2475	Breastfeeding, complementary feeding, formula	-	-	-	Questionnaire	-
Generation R Study	Cohort study	52 [51,52,54,55,56,57,58,59,62,64,65,66,67,68,70,71,72,73,78,79,81,86,87,89,90,93,95,96,97,98,99,100,151,152,153,154,155,156,157,158,159,160,161,162,163,164,165,166,167,168,169,170]	2001–2006	444–7857	Bread, breastfeeding, complementary feeding, composite dishes, confectionary, dairy products, eggs, fat, fish, formula, fruit, legumes, meat, non-alcoholic drinks, pasta, rice, potatoes, sauces, snacks, soup, soya and diet products, vegetables	Vitamin D	Energy, beta-carotene, calcium, carbohydrates, fat, fiber, magnesium, phosphorus, potassium, protein, sodium	25 (OH)D	FFQ ^3^, questionnaire	Umbilical cord blood
IROSTAT Study	Case-control study	2 [105,171]	2011–2012	313–351	-	-	-	Ferritin	-	Serum
KOALA	Cohort study	8 [106,108,109,110,111,172,173,174]	2000–2010	521–2818	Bread, breastfeeding, formula, fruit, non-alcoholic drinks, snacks, vegetables	Vitamin D, multivitamin	-	-	FFQ, questionnaire	-
LOOZ	Cohort study	1 [175]	2003	600	Breastfeeding	-	-	-	Questionnaire	-
MEFAB	Cohort study	1 [113]	1989–1995	292	Breastfeeding	-	-	-	Questionnaire	-
‘Peiling melkvoeding van zuigelingen’	National survey	1 [116]	2015	1740	Breastfeeding, formula	-	-	-	Questionnaire	-
PIAMA	Cohort study	5 [117,118,176,177,178]	1996–1997	3684–3963	Breastfeeding, complementary feeding, formula	-	-	-	FFQ, questionnaire	-
Sophia Pluto Study	Cohort study	1 [179]	2013-ongoing	197	Breastfeeding	-	-	-	Questionnaire	-
VoorZorg	RCT	1[180]	2007–2009	223	Breastfeeding	-	-	-	Questionnaire	-
WHISTLER	Cohort study	3 [181,182,183]	2001–2012	1056	Breastfeeding, formula	-	-	-	Questionnaire	-
ZOOG	Cross-sectional study	1 [184]	-	9	Breastfeeding	Vitamin D	-	-	Questionnaire	
Akkermans et al.	Cross-sectional study	1 [185]	2012–2014	45	-	Iron and vitamin D	Iron, vitamin D	Ferritin	Questionnaire	Blood
Barends et al.	RCT	1 [186]	2010–2011	71	-	-	Energy, carbohydrates, fat, fiber, protein	-	3d food record ^3^	-
Beijers et al.	Cohort study	1 [187]	-	193	Breastfeeding	-	-	-	Questionnaire	-
Belderbos et al.	Cohort study	2 [124,188]	2006–2009	156–291	Breastfeeding	Vitamin D	-	25(OH)D	Questionnaire	Umbilical cord
Biesbroek et al.	RCT	1 [189]	2005–2006	202	Breastfeeding	-	-	-	Questionnaire	-
Bosch et al.	Cohort study	1 [190]	-	112	Breastfeeding	-	-	-	Questionnaire	-
Bulk-Bunschoten et al.	Cohort study	1 [191]	1998	4438	Breastfeeding		-	-	Recall	-
Diepeveen et al.	Case-control study	1 [125]	-	253	Breastfeeding	-	-	-	Questionnaire	-
Dirix et al.	Case-control study	1 [126]	-	90	-	-	-	Fatty acids	Questionnaire	Umbilical cord
Groen-Blokhuis et al.	Cohort study	1 [192]	-	-	Breastfeeding	-	-	-	Questionnaire	-
Hogeman et al.	Cohort study	1 [193]	2006	74	Breastfeeding	Vitamin D	-	25(OH)D, calcium	Questionnaire	Serum
Hopman et al.	Cross-sectional study	1 [194]	2008	25	-	-	-	-	FFQ ^3^	-
Obermann-Borst *et al.*	Cohort study	1 [130]	2003–2007	120	Breastfeeding	Folic acid	-	-	Questionnaire	-
Boon et al. (RIKILT, RIVM, TNO)	Surveys	1 [195]	1987–2002	643	Bread, (breakfast) cereals, cakes and biscuits, dairy products, fish, fruit, meat, non-alcoholic beverages, nuts and nut products, potatoes, vegetables	-	-	-	2d food record, FFQ, 1d food record & duplicate portion	-
Verkaik-Kloosterman et al. (RIVM)	National surveys	1 [196]	1997–1998,2005–2006	254	-	-	Iodine	-	2d food record	-
Verkaik-Kloosterman et al. (RIVM)	Cross-sectional study	1 [197]	2002	scenario analysis	Formula	Vitamin D	Energy	-	2d food record	-
Verkaik-Kloosterman (RIVM)	Cross-sectional study, national survey	1 [198]	2002;2005–2006	-	-	Retinol	Retinol	-	2d food record	-
Uijterschout et al.	Cross-sectional study	1 [199]	2011–2012	351	Breastfeeding	-	-	Ferritin	Questionnaire	Blood
Van Eijsden et al.	Case-control study	1 [200]	2009–2010	286	Bread, (breakfast) cereals, breastfeeding, complementary feeding, confectionery, dairy products, fruit, vegetables	-	-	-	Recall	-
Van Goor et al.	RCT	1 [134]	2004–2006	36	Breastfeeding	-	-	-	Questionnaire	RBC
Weernink et al.	Case-control study	1 [137]	2009–2010	548	Breastfeeding, formula	Vitamin D	-	-	Questionnaire	-
Weijs et al.	Cohort study	1 [201]	2001	63	Breastfeeding, formula	-	-	-	2d food record	-

^1^ as multiple references were available, the minimum and maximum number of the reported sample size are stated; ^2^ may include nutrient intake from supplements; ^3^ dietary assessment method was reported as validated; 25(OH)D = 25-hydroxyvitamin D; FFQ = food frequency questionnaire; RCT = randomized controlled trial.

**Table 3 nutrients-11-00860-t003:** Overview of reported Dutch food consumption data and study characteristics, for pregnant women and children up to two years, published since 2008.

Food Consumption	Pregnant Women					Children up to 2 Years				
	Number of Articles	Number of Individual Studies	Number of Participants	Year of Assessment	Method	Number of Articles	Number of Individual Studies	Number of Participants	Year of Assessment	Method
Alcohol	75	18	59–7890	1989–2016	FFQ, Q	1	1	517	2012–2016	2d recall, 2d FR
Bread	2	1	847–3207	2012–2013	FFQ	7	5	286–2420	2009–2016	FFQ, (2d) recall, 2d FR, 1d FR & DP
Breakfast-cereals	1	1	3207	2001–2006	FFQ	5	4	286–1526	1987–2014	(2d) recall, 2d FR, 1d FR & DP
Breastfeeding	1	1	8	2008	Q	96	28	49–7210	1989–2015	FFQ, Q, recall, 2d FR
Confectionary/sweets	1	1	3207	2001–2006	-	10	8	286–2420	1987–2014	FFQ, (2d) recall, 2d FR, 1d FR & DP
Complementary feeding	0	0	-	-	-	23	5	286–7857	2003–2010	FFQ, Q
Composite dishes	0	0	-	-	-	5	3	517–2420	2003–2014	FFQ, (2d) recall, 2d FR
Condiments	2	1	847–3207	2001–2006	FFQ	1	1	517	2012–2014	2d recall, 2d FR
Dairy products	5	3	548–3963	1996–2010	FFQ, Q	8	5	517–2420	1987–2014	FFQ, (2d) recall, 2d FR, 1d FR & DP
Eggs	3	2	847–3963	1996–2006	FFQ, Q	5	3	517–2420	2003–2014	FFQ, (2d) recall, 2d FR
Fat	2	1	847–3207	2001–2006	FFQ	5	3	517–2420	2003–2014	FFQ, (2d) recall, 2d FR
Fish	14	5	36–7210	1996–2012	FFQ, Q	7	4	517–7210	1987–2014	FFQ, (2d) recall, 2d FR, 1d FR & DP
Formula	0	0	-	-	-	15	12	63–3629	1996–2015	FFQ, Q, 2d FR
Fruit	6	4	455–6021	1996–2012	FFQ, Q	11	7	286–3624	1987–2014	FFQ, Q, (2d) recall, 2d FR, 1d FR & DP
Legumes	2	1	847–3207	2001–2006	FFQ	5	3	517–2420	2002–2008	FFQ, (2d) recall, 2d FR
Meat	2	1	847–3207	2001–2006	FFQ	6	4	517–2420	2003–2014	FFQ, (2d) recall, 2d FR, 1d FR & DP
Non-alcoholic drinks	3	1	847–3312	2001–2006	FFQ	7	5	517–2420	1987–2014	FFQ, (2d) recall, 2d FR, 1d FR & DP
Nuts and nut products	2	2	847–3963	1996–2006	FFQ, Q	4	3	517–1526	1987–2014	Q, (2d) recall, 2d FR, 1d FR & DP
Pasta, rice	2	1	847–3207	2001–2006	FFQ	2	1	2420	2003–2006	FFQ
Potatoes	2	1	847–3207	2001–2006	FFQ	6	4	517–2420	1987–2014	FFQ, (2d) recall, 2d FR, 1d FR & DP
Sauces	1	1	3207	2001–2006	FFQ	5	3	517–2420	2003–2014	FFQ, (2d) recall, 2d FR
Snacks	1	1	847	2001–2006	FFQ	5	3	517–2420	2003–2014	FFQ, (2d) recall, 2d FR
Soup	1	1	3207	2001–2006	FFQ	5	3	517–2420	2003–2014	FFQ, (2d) recall, 2d FR
Soya and diet products	1	1	3207	2001–2006	FFQ	4	2	939–2420	2003–2014	FFQ, 2d FR
Vegetables	7	4	455–6021	1996–2012	FFQ, Q	10	7	286–3624	1987–2014	FFQ, Q, (2d) recall, 2d FR, 1d FR & DP

DP = duplo portion; FFQ = food frequency questionnaire; FR = food record; Q = questionnaire.

**Table 4 nutrients-11-00860-t004:** Overview of reported nutrient intake data and study characteristics, for pregnant women and children up to two years, published since 2008.

Nutrient Intake	Pregnant Women					Children up to 2 Years				
	Number of Articles	Number of Individual Studies	Number of Participants	Year of Assessment	Method	Number of Articles	Number of Individual Studies	Number of Participants	Year of Assessment	Method
energy	28	5	46–7890	1999–2009	FFQ, Q	20	4	71–5225	1999–2014	FFQ, 2d FR, 3d FR & semi-weighted
carbohydrates	10	2	46–7346	2001–2006	FFQ, Q	10	3	71–3610	2001–2014	FFQ, 2d FR, 3d FR & semi-weighted
fat	12	4	46–7346	2001–2009	FFQ, Q	13	3	71–4830	2002–2014	FFQ, 2d FR, 3d FR & semi-weighted
fiber	3	1	2420–3207	2002–2006	FFQ	7	3	71–2420	2002–2014	FFQ, 2d FR, 3d FR & semi-weighted
protein	13	2	46–7346	2001–2009	FFQ, Q	15	3	71–4637	2003–2014	FFQ, 2d FR, 3d FR & semi-weighted
thiamin	0	0	-	-	-	2	1	939–1526	2011–2014	2d FR
riboflavin	2	1	251–324	2003–2006	FFQ	2	1	939–1526	2011–2014	2d FR
niacin/nicotinamide	1	1	324	2003–2006	FFQ	2	1	939–1526	2011–2014	2d FR
vitamin A/retinol	1	1	46	2006–2009	FFQ	4	3	939–1526	2002–2014	2d FR
beta-carotene	0	0	-	-	-	1	1	2044	2003	FFQ
pyridoxine	0	0	-	-	-	2	1	939–1526	2011-2014	2d FR
cobalamin	0	0	-	-	-	2	1	939–1526	2011–2014	2d FR
folate	2	1	251–324	2003–2006	FFQ	2	1	939–1526	2011–2014	2d FR
vitamin C	0	0	-	-	-	2	1	939–1526	2011–2014	2d FR
vitamin D	0	0	-	-	-	3	2	45–1526	2011–2014	Q, 2d FR
vitamin E	0	0	-	-	-	2	1	939–1526	2011–2014	2d FR
calcium	2	1	2819–2683	2002–2006	FFQ	3	2	939–2850	2003–2014	FFQ, 2d FR
copper	0	0	-	-	-	2	1	939–1526	2011–2014	2d FR
iodine	0	0	-	-	-	3	2	254–1526	1997–2014	2d FR
iron	1	1	2863	2002–2006	FFQ	3	2	45–1526	2011–2014	Q, 2d FR
magnesium	1	1	2819	2002–2006	FFQ	3	2	939–2850	2003–2014	FFQ, 2d FR
phosphorus	1	1	2819	2002–2006	FFQ	3	2	939–2850	2003–2014	FFQ, 2d FR
potassium	0	0	-	-	-	3	2	939–2850	2003–2014	FFQ, 2d FR
selenium	0	0	-	-	-	2	1	939–1526	2011–2014	2d FR
sodium	2	1	2863–6215	2002–2006	FFQ	3	2	939–2968	2002–2014	FFQ, 2d FR
zinc	0	0	-	-	-	2	1	939–1526	2011–2014	2d FR
caffeine	5	2	3439–7890	2001–2006	FFQ, Q	0	0	-	-	-
water	0	0	-	-	-	2	1	939–1526	2011–2014	2d FR

FFQ = food frequency questionnaire; FR = food record; Q = questionnaire.

**Table 5 nutrients-11-00860-t005:** Overview of reported biochemical nutrient status data and study characteristics, for pregnant women and children up to two years, published since 2008.

Biochemical Nutrient Intake Status	Pregnant Women					Children up to 2 Years				
	Number of Articles	Number of Individual Studies	Number of Participants	Year of Assessment	Method	Number of Articles	Number of Individual Studies	Number of Participants	Year of Assessment	Method
fatty acids	9	6	36–4830	1989–2006	blood, plasma, RBC	1	1	90	1989–2006	UC phospholipids
cobalamin	8	2	847–4389	2001–2014	blood, plasma, serum	0	0	-	-	-
folate	26	3	420–4389	2001–2014	blood, plasma, serum	0	0	-	-	-
homocysteine	15	2	420–3207	2001–2014	blood, plasma, serum	0	0	-	-	-
25-hydroxyvitamin D	14	3	1356–7256	2000–2006	blood, plasma, serum	6	3	74–5294	2002–2009	blood, plasma, serum, UC blood
retinol	1	1	46	2006–2009	serum	0	0	-	-	-
calcium	0	0	-	-	-	1	1	74	2006	serum
ferritin	1	1	31	2009–2011	serum	4	3	45–351	2011–2014	blood, serum
iodine	1	1	1525	2002–2006	single spot urine	0	0	-	-	-
iron	1	1	31	2009–2011	serum	0	0	-	-	-

RBC = red blood cells; UM = umbilical cord.

**Table 6 nutrients-11-00860-t006:** Overview of reported supplement data and study characteristics, for pregnant women and children up to two years, published since 2008.

Supplement Use	Pregnant Women					Children up to 2 Years				
	Number of Articles	Number of Individual Studies	Number of Participants	Year of Assessment	Method	Number of Articles	Number of Individual Studies	Number of Participants	Year of Assessment	Method
Vitamin A	1	1	3963	1996	Q	0	0	-	-	-
B Vitamins	1	1	3963	1996	Q	0	0	-	-	-
Vitamin C	1	1	3963	1996	Q	0	0	-	-	-
Vitamin D	5	3	156–7256	2002–2010	Q	20	10	9–5322	2002–2014	FFQ, Q, 2d FR
Folic acid	56	13	46–7890	1996–2016	Q	2	2	517–2911	2001–2014	Q
Multivitamin	9	3	46–3963	1996–2014	Q	2	2	517–1356	2002–2014	Q
Calcium	1	1	3963	1996	Q	1	1	517	2012–2014	Q
Iron	3	2	313–3963	1996–2005	Q	1	1	1356	2012–2014	Q
Fish oil	2	2	2622–3254	2000–2004	FFQ, Q	0	0	-	-	-

FFQ = food frequency questionnaire; Q = questionnaire.

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
