# Peer review of "Food Consumption, Nutrient Intake and Status during the First 1000 Days of Life in The Netherlands: A Systematic Review"

_nutrients, 2019, doi:10.3390/nu11040860_

Round 1

Reviewer 1 Report

This review is of interest. Yet it can be further improved.

First, it would be useful to provide further details on the rationale of selecting 200 as the starting point for the literature review: sentence 93-95 arenot clear. 

Second, with regard to the presentation of the result, it might be good to also incude the followings: 

- Studies characteristics: dietary assessment methods, location type of design (RCT, longitudinal, other)

- Risks of biais: studies quality

- Participant characteristics: age of women, parity, gender of children and age, etc.

An then, finish the results section with the description of outcomes (food, consumption, nutrient intake, etc.). It might be good to present data in children based on their breastfeeding status. 

In Table 1, it would be useful to have a column on the type of the study`s design. 

The discussion should be reviewed accordingly and be more organized.

The English language of the text requires major revisions. 

Author Response

Please see the uploaded pdf document for our response to the comments of reviewer I.

Reviewer 2 Report

Dear Editor,

Thank you for the opportunity to review the manuscript titled "Food consumption, nutrient intake, and status during the first 1000 days of life in the Netherlands: a systematic review". I read the manuscript with much enthusiasm as this manuscript addresses an essential gap in the area of food and nutrient consumption. Although multiple articles are available on food nutrient intakes of different groups of populations, a comprehensive review could provide an overall view and identify the missing links in this area. 

I have a few suggestions to the authors:

Line number 43. Women in affluent countries are often low in their iodine intakes also. Addition of Iodine to the list of nutrients mentioned would be valuable.

Line number 45: The intake of Omega 3 fatty acids are essential as well. The consumption of Omega 3 fatty acids through supplement use reported being higher among populations from the West.

Line number 89-90: PRISMA is a reporting guideline for systematic reviews and not a guideline for conducting systematic reviews. The statement needs to be corrected.

Authors have not mentioned whether they had used any software for data extraction and analysis. Please state a line in this regards.
Authors have mentioned that two independent researchers completed the abstract, title selection. It would be good to indicate how authors addressed the disagreements when two of the researchers could not conclude on a particular study selection. 
How did the authors assess the quality of the studies included in the review? Even if it is a narrative review, it is relevant to have the quality of data evaluated before deriving the conclusions. Quality of systematic reviews increases with the appropriate use of the defined protocols. If the quality of the included studies is not assessed, this could lead to misinterpretation of data.
Authors may include the types of studies they have used- for example, observational, cross-sectional, cohort or any other- of the studies they have included in the systematic review.

Author Response

Please see the uploaded pdf, with our response to the comments of reviewer II

Round 2

Reviewer 1 Report

I agree with all changed provided by authors. 

Reviewer 2 Report

Greetings!

Authors have addressed the reviewer's comments satisfactorily.